# Scalable focused ion beam creation of nearly lifetime-limited single quantum emitters in diamond nanostructures

Tim Schröder[1,*,†], Matthew E. Trusheim[1,*], Michael Walsh[1,*], Luozhou Li[1], Jiabao Zheng[1,†], Marco Schukraft[1], Alp Sipahigil[2], Ruffin E. Evans[2], Denis D. Sukachev[2,†], Christian T. Nguyen[2], Jose L. Pacheco[3], Ryan M. Camacho[3], Edward S. Bielejec[3], Mikhail D. Lukin[2] & Dirk Englund[1]

The controlled creation of defect centre—nanocavity systems is one of the outstanding challenges for efficiently interfacing spin quantum memories with photons for photon-based entanglement operations in a quantum network. Here we demonstrate direct, maskless creation of atom-like single silicon vacancy (SiV) centres in diamond nanostructures via focused ion beam implantation with $\sim 32$ nm lateral precision and $<50$ nm positioning accuracy relative to a nanocavity. We determine the Si+ ion to SiV centre conversion yield to be $\sim 2.5\%$ and observe a 10-fold conversion yield increase by additional electron irradiation. Low-temperature spectroscopy reveals inhomogeneously broadened ensemble emission linewidths of $\sim 51$ GHz and close to lifetime-limited single-emitter transition linewidths down to $126 \pm 13$ MHz corresponding to $\sim 1.4$ times the natural linewidth. This method for the targeted generation of nearly transform-limited quantum emitters should facilitate the development of scalable solid-state quantum information processors.

[1] Department of Electrical Engineering and Computer Science, Massachusetts Institute of Technology, Cambridge, Massachusetts 02139, USA. [2] Department of Physics, Harvard University, 17 Oxford Street, Cambridge, Massachusetts 02138, USA. [3] Sandia National Laboratories, Albuquerque, New Mexico 87185, USA. * These authors contributed equally to this work. † Present addresses: Niels Bohr Institute, University of Copenhagen, Denmark (T.S.); Columbia University, New York, New York 10027, USA (J.Z.); Russian Quantum Center and P.N. Lebedev Physical Institute, Moscow 143025, Russia (D.D.S.). Correspondence and requests for materials should be addressed to T.S. (email: schroder@nbi.ku.dk) or to D.E. (email: englund@mit.edu).

A central goal in semiconductor quantum optics is to devise efficient interfaces between photons and atom-like quantum emitters for applications including quantum memories, single-photon sources and nonlinearities at the level of single quanta. Many approaches have been investigated for positioning emitters relative to the mode-maximum of nanophotonic devices with the necessary subwavelength-scale precision, including fabrication of nanostructures around pre-localized or site-controlled semiconductor quantum dots[1–5] or diamond defect centres[6], or implantation of ions for defect centre creation in nanostructures concomitant with the nanofabrication[7,8]. However, these approaches have not allowed high-throughput post-fabrication creation of quantum emitters with nearly indistinguishable emission in nanophotonic structures already fabricated and evaluated; such an approach greatly simplifies the design and fabrication process and improves the yield of coupled emitter–nanostructure systems.

Unlike quantum emitters such as molecules or quantum dots, diamond defect centres can be created through ion implantation and subsequent annealing[9,10], enabling direct control of the centre depth via the ion energy. Lateral control has been demonstrated through the use of nanofabricated implantation masks[11–16], which have been employed for colour centre creation relative to optical structures through atomic force microscope (AFM) mask alignment[6], and combined implantation/nanostructure masking[7,8]. Implantation through a pierced AFM tip[6] does not require modification of the fabrication process and allows for implantation after fabrication and evaluation of these structures. However, the process is time-consuming, requires special AFM tips and can lead to reduced positioning precision by collisions with mask walls. As an alternative, focused ion beam (FIB) implantation of ions, for example, nitrogen[17] and silicon[18], can greatly simplify the implantation process by eliminating the need of a nanofabricated mask. Similar to a scanning electron microscope, an ion beam can be precisely scanned, enabling lateral positioning accuracy at the nanometre scale and 'direct writing' into tens of thousands of structures with high throughput.

The silicon vacancy (SiV) belongs to a group of colour centres in diamond that has emerged as promising single-photon emitters and spin-based quantum memories. Among the many diamond-based fluorescent defects that have been investigated[19], the SiV centre[20–23] is exceptional in generating nearly lifetime-limited photons with a high Debye-Waller factor of 0.79 (ref. 24) and low spectral diffusion due to a vanishing permanent electric dipole moment in an unstrained lattice[25,26]. These favourable optical properties have notably enabled two-photon quantum interference between distant SiV centres[25,27] and entanglement of two SiV centres coupled to the same waveguide[28]. In addition, the SiV has electronic and nuclear spin degrees of freedom that could enable long-lived, optically accessible quantum memories[29–31].

Here we introduce a method for positioning emitters relative to the mode-maximum of nanophotonic devices: direct FIB implantation of Si ions into diamond photonic structures. This post-fabrication approach to quantum emitter generation achieves nanometre-scale positioning accuracy and creates SiV centres with optical transition linewidths comparable to the best 'naturally' growth-incorporated SiV reported[27]. The approach allows Si implantation into $\sim 2 \times 10^4$ sites per second, which allows creation of millions of emitters across a wafer-scale sample. We also show that additional post-implantation electron irradiation and annealing creates an order of magnitude enhancement in Si to SiV conversion yield. By repeated cycles of Si implantation and optical characterization, this approach promises nanostructures with precisely one SiV emitter per desired location. Finally, we demonstrate and evaluate the

site-targeted creation of SiVs in pre-fabricated diamond photonic crystal nanocavities. The ability to implant quantum emitters with high spatial resolution and yield opens the door to the reliable fabrication of efficient light–matter interfaces based on semiconductor defects coupled to nanophotonic devices.

## Results

**Spatial precision of SiV creation.** As outlined in Fig. 1, the fabrication approach introduced here relies on Si implantation in a custom-built 100 kV FIB nanoImplanter (A&D FIB100nI) system (Methods) and subsequent high-temperature annealing to create SiV centres. The nanoImplanter uses field emission to create a tightly focused ion beam down to a minimum spot size of < 10 nm from a variety of liquid metal alloy ion sources (Methods). For the experiments described here, we used an Si beam with a typical spot size of < 40 nm into commercially available high-purity chemical vapour deposition diamond substrates (Element6). After implantation, we performed high-temperature annealing and surface preparation steps to convert implanted Si ions to SiVs (Methods).

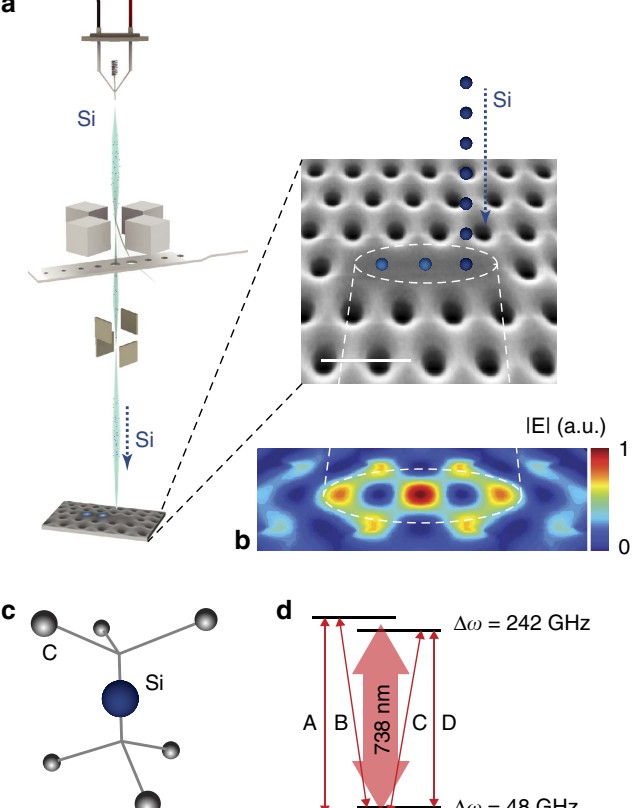

**Figure 1 | Targeted Si ion implantation into diamond and SiV defect properties.** (**a**) Illustration of targeted ion implantation. Si ions are precisely positioned into diamond nanostructures via a FIB. The zoom-in shows a scanning electron micrograph of a L3 photonic crystal cavity patterned into a diamond thin film. Scale bar, 500 nm; Si is silicon. (**b**) Intensity distribution of the fundamental L3 cavity mode with three Si target positions: the three mode-maxima along the centre of the cavity are indicated by the dashed circle. The central mode peak is the global maximum. (**c**) Atomic structure of a SiV defect centre in diamond. Si represents an interstitial Si atom between a split vacancy along the <111> lattice orientation and C the diamond lattice carbon atoms. (**d**) Simplified energy-level diagram of the negatively charged SiV indicating the four main transitions A, B, C and D[26]. $\Delta\omega$ is the energy splitting of the two levels within the doublets.

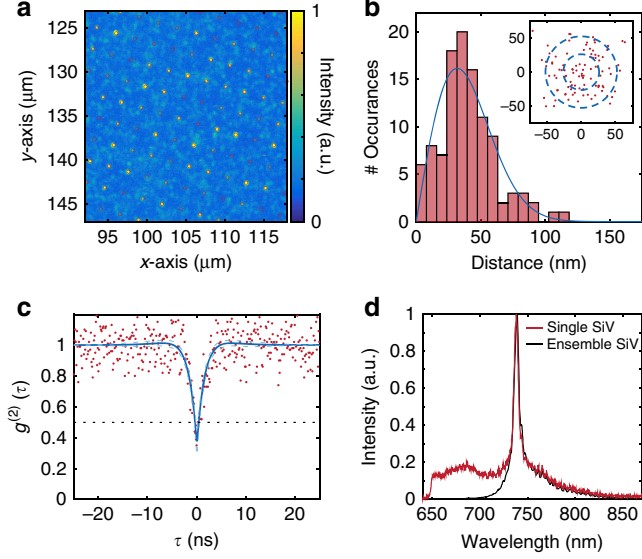

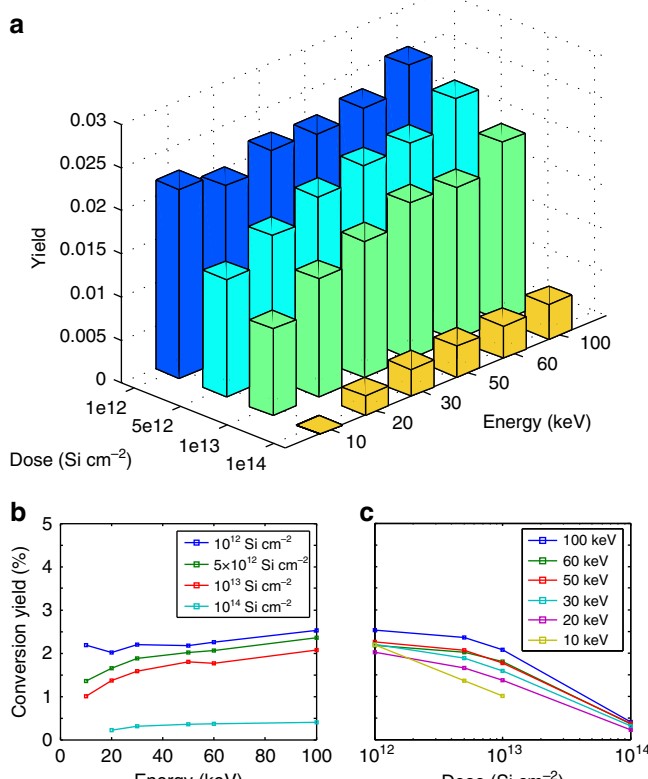

**Figure 2 | Spatial precision of SiV creation.** (**a**) Confocal scan of SiV centre array. Sites are separated by 2.14 µm. Overlaid are regular grid points from an aberration-corrected reference lattice. (**b**) Analysis of implantation precision. We determine the 2D position uncertainty of the created SiV be $40 \pm 20$ nm. Blue curve: fit to Rayleigh distribution. Inset: Scatter plot of created single SiV sites relative to their grid points with one and two $\sigma$ guides to the eye, where the radius $\sigma = 26$ nm corresponds to the expected implantation s.d. resulting from the combination of beam width and implant straggle. (**c**) Normalized second-order autocorrelation function of a single SiV with $g^{(2)}(0) = 0.38 \pm 0.09$. Red points indicate data (without background subtraction), and the blue line is a fit to the function $1 - A \cdot \exp(-|\tau/t_1|) + B \cdot \exp(-|\tau/t_2|)$. The black dashed line indicates $g^{(2)}(\tau) = 0.5$, while the blue dashed lines indicate the 95% confidence interval on the fit. (**d**) Ensemble (black) and single-emitter (red) SiV room temperature fluorescence spectra. The characteristic zero-phonon line at 737 nm is prominent.

We characterized the resulting SiV arrays at room temperature through confocal fluorescence microscopy in a home-built set-up (Methods). Figure 2a shows a scan of a square array of SiV implantation sites with lattice spacing of 2.14 µm across a $30 \times 30 \, \mu m^2$ write field, created via a single-point exposure from the Si beam. Room temperature spectral measurements in a dense region containing many centres (Fig. 2d, blue curve) showed a zero-phonon line (ZPL) with an inhomogeneous linewidth of $\sim 5$ nm centred around 738.3 nm, characteristic of the SiV centre. We subsequently identified single SiVs through second-order correlation measurements. For instance, Fig. 2c shows photon antibunching for a SiV with an observed count rate of 30 kcts s$^{-1}$ collected via an oil immersion (numerical aperture (NA) of 1.3) objective into a single-mode fibre under 20 mW of 532 nm pump power. The red line in Fig. 2d shows the single-emitter fluorescence spectrum at room temperature, which is very similar in shape and linewidth to the inhomogeneous spectrum. At room temperature, these lines are broadened by phonon processes and not limited by inhomogeneity between different SiV centres[32].

To determine the spatial precision of creating SiV with our method, we analysed their distribution relative to the implantation lattice grid. Figure 2b shows the distance of each imaged implanted single SiV from the ideal lattice site, resulting in a $\chi$-distribution with a mean separation in R of $\sigma = 40 \pm 20$ nm and underlying lateral $(x, y)$ distributions with zero mean and s.d.'s of 32 nm. These measured values agree well with the expected precision of 26 nm calculated by the addition in quadrature of the

**Figure 3 | Si ion to SiV conversion yield.** (**a**) Si to SiV conversion yield for varying implantation ion energies and doses. The conversion yield was determined by calibrating array intensities (Fig. 2) with the determined averaged single SiV photon count rate. Si conversion yield as function of (**b**) energy and (**c**) dose. The lines are guides to the eye.

uncertainties arising from the nominal 40 nm FHWM beam size and 19 nm lateral implantation straggle.

**Creation yield of SiV.** To determine the conversion yield of implanted Si ions to SiV centres, we swept the implantation dose logarithmically from $10^{12}$ to $10^{14}$ Si cm$^{-2}$, and the implantation energy linearly from 10 to 100 keV. The dose and energy determine the number and depth of vacancies created during the implantation process, with increased energy resulting in more vacancies at increased depth. The vacancy density affects the probability that a Si defect captures a diffusing vacancy and converts to SiV during annealing, which is a proposed mechanism for SiV formation[33]. To estimate the yield, we measured the fluorescence intensity across a $10 \times 10 \, \mu m^2$ region of constant implantation dose and energy, and normalized to the average single-emitter intensity and implanted ion number. Figure 3a summarizes the yield measurements. Yield increases as a function of energy (Fig. 3b), which is expected for a vacancy-limited SiV creation process, up to 2.5% for the highest-energy 100 keV ions with a dose of $\sim 10^{12}$ cm$^{-2}$. These measurements indicated a decreasing yield as a function of dose (Fig. 3c). We attribute this to an accumulation of charged defects in the diamond lattice that lead to ionization, similarly to what was observed in nitrogen vacancy (NV) centres[34]. Alternatively, reduced yield could result from lattice damage that accumulates in the form of multivacancy defects as the diamond lattice approaches the graphitization threshold, a phenomenon that has been observed in similar experiments with NV centres[35].

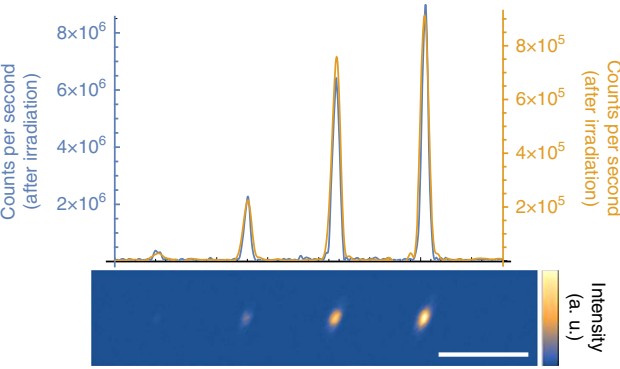

**Figure 4 | Electron co-implantation.** After electron irradiation and subsequent annealing we observe a 10-fold increase in fluorescence intensity at the implantation positions of Si ions (lower inset). The Si ion doses were 500, 2,000, 5,000 and 10,000 ions per spot. The yellow line plot through the fluorescence maximum of the image indicates the intensity before electron irradiation, and the blue line after irradiation. Scale bar, 5 μm.

Irradiating diamond with high-energy electrons can also improve the conversion yield of vacancy-related colour centres[36,37]. Electron irradiation at high energies > 170 keV (ref. 38) can displace carbon atoms and create additional vacancies, which allows for larger conversion efficiency of implanted ions into vacancy-related colour centres. To verify these experiments with the SiV centre, we first created a reference sample by implanting four spots with silicon ions in increasing doses of 500, 2,000, 5,000 and 10,000 ions per spot into bulk diamond with an implantation energy of 100 keV, corresponding to an implantation depth of ~68 nm. After annealing this sample at 1,200 °C to activate SiVs[39], a scanning confocal fluorescence image was taken by exciting these spots simultaneously with ~10 mW of both 520 nm (Thorlabs LP520-SF15) and 700 nm (Thorlabs LP705-SF15) laser light, and collecting light into a single-mode fibre through a 10 nm bandpass filter (Semrock FF01-740/13) around 737 nm (Fig. 4, yellow line). After this reference measurement, we irradiated the sample with 1.5 MeV electrons with a total fluence of ~$10^{17}$ cm$^{-2}$. After another annealing step, a second fluorescence image was taken with the same set-up and it was verified by spectral measurements (Horiba iHR 550 with Synapse CCD) that indeed only the SiV typical peak at 737 nm was detected. In the second measurement, we observed increased fluorescence for all four spots by a factor of ~10 (Fig. 4, blue line), corresponding to a final conversion yield of ~20%. This result is consistent with previous observations in Si-doped diamond samples[33], supporting our interpretation that the conversion efficiency of FIB implantation is limited by the vacancy density in the diamond.

**Optical and coherence properties of SiV at cryogenic temperatures.** We next describe the implanted SiV centres' low-temperature spectral properties. Photoluminescence spectral measurements were performed in a home-built confocal cryostat set-up (Methods). The inhomogeneous distribution of SiV transitions at 18 K is plotted in Fig. 5a with a full width at half maximum (FWHM) of ~0.642 nm (~51 GHz). We then performed photoluminescence excitation (PLE) measurements determine the linewidths of individual SiVs below the spectrometer limit (Methods). We determined an average single-emitter transition linewidth of 200 ± 15 MHz from a sample of 10 SiV implanted at 100 keV with individually resolvable transitions. The narrowest observed transition, shown in Fig. 5b, had a linewidth of

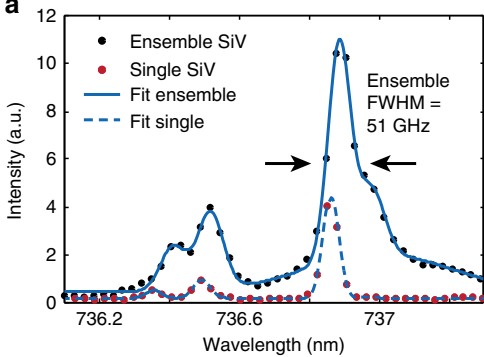

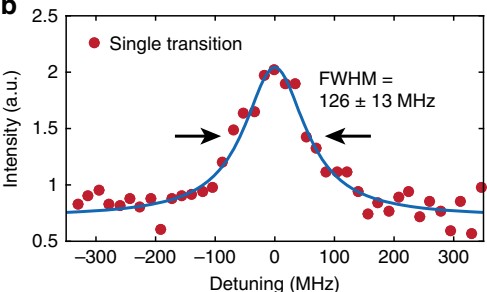

**Figure 5 | Optical linewidth and coherence properties of SiV.** (**a**) Cryogenic spectra (<18 K) of a single SiV (red circles) and an ensemble (black circles). The four SiV transitions (Fig. 1) as well as the phonon sideband are each fitted with a Gaussian function. The single SiV linewidths are spectrometer-limited (FWHM = ~34 GHz). For the ensemble, we determine an inhomogeneous broadening as low as ~51 GHz (FWHM). The wavelength values are slightly blue-shifted because of an offset relative to an absolute wavelength reference by ~0.1 nm. (**b**) Cryogenic (4 K) photoluminescence excitation measurement of the narrowest observed single SiV transition with a linewidth of 126 ± 13 MHz (FWHM, error estimation: 95% confidence interval) determined with a Lorentzian fit function. This linewidth of an implanted SiV is equal, within error, to the narrowest natural SiV linewidth measured to date.

126 ± 13 MHz, which is within a factor of 1.4 of the lifetime limit $\gamma = (2\pi \times 1.7 \text{ ns})^{-1} = 94$ MHz for a typical fluorescence lifetime of 1.7 ns (ref. 25), and equivalent to the narrowest lines observed in natural SiVs to date[27,40].

**Direct SiV creation in an optical nanocavity.** Finally, we demonstrated the targeted implantation and subsequent creation of SiV centres inside diamond nanostructures. We first fabricated two-dimensional (2D) photonic crystal nanocavities into a ~200-nm-thick diamond membrane through oxygen reactive ion etching[41,42]. We then used the FIB system to target Si ions into the mode-maxima of the photonic crystal cavities. In the case of L3 cavities, we targeted the three mode-maxima individually (Fig. 1b). The Si ion beam was aligned to the cavity through secondary-electron imaging of pre-fabricated alignment markers on the sample (Fig. 6b, Methods). We targeted the cavity mode-maxima with 160 keV ions for an average of 1.8 SiVs per cavity (Methods). After performing the processing steps described in Methods, we observed about one SiV per cavity implantation spot with spectrometer-limited (<34 GHz) ZPL linewidths. To determine the position of a single SiV relative to the cavity, we performed a spectrally resolved photoluminescence confocal scan (Fig. 6d,e). This measurement allows comparison between the photonic crystal cavity location, determined by Raman scattering, and the SiV location, determined from the emitted fluorescence at the SiV ZPL. By fitting the measured emission patterns to 2D

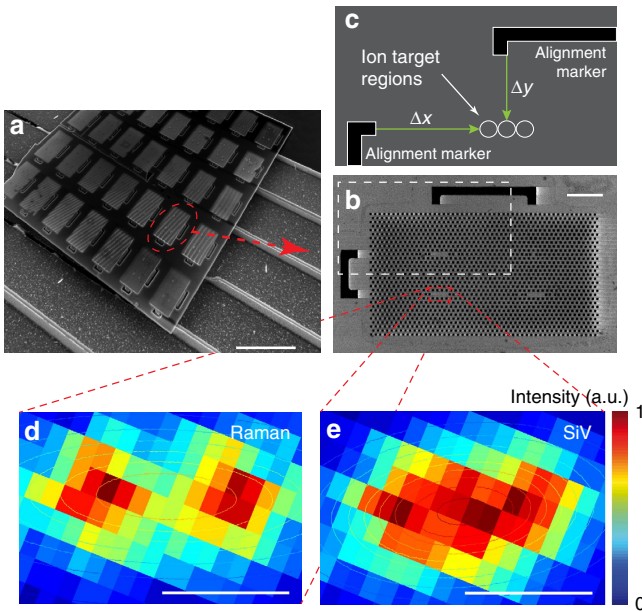

**Figure 6 | SiV creation in photonic nanocavity.** (**a**) SEM of example PhC cavity sample. Scale bar, 10 μm. (**b**) Close-up SEM of example PhC lattice with four cavities. The white dashed rectangle indicates the area illustrated in **c**. Scale bar, 2 μm. (**c**) Illustration of targeting relative to alignment markers (black) with an ion spot size down to <40 nm. The white circles (not to scale for visibility) indicate the three L3 cavity mode-maxima (Fig. 1). To determine the SiV-positioning accuracy relative to the mode-maxima, we performed spectrally resolving fluorescence scans. At each pixel in **d,e**, we recorded a spectrum including the Raman signal (**d**), the SiV fluorescence (**e**) and the cavity resonances (not displayed). (**d**) Intensity x–y plot of the diamond Raman signal at 572.8 nm. (**e**) Intensity x–y plot of SiV emission at 736.9 nm. By fitting a 2D Gaussian function to the intensity distribution, we determined the distance between the centre of the cavity and the SiV fluorescence, the effective positioning accuracy, to 48(21) nm, with error estimation of one s.d. Scale bar in **d,e**, 0.5 μm.

Gaussians, we estimate a relative positioning accuracy of $48 \pm 21$ nm. This is close to the limit set by the combination of beam size and implantation straggle ($\sim 52$ nm), with very low offset of 4 nm in the $x$ direction and a main offset of 48 nm in the $y$ direction.

## Discussion

While we have demonstrated targeted creation of high-quality SiVs through FIB, there are several avenues for improvement. The stochastic nature of the SiV creation process, characterized by a mean yield of $\eta$, prevents the generation of exactly one emitter with high yield[43]. One solution is to implant a low dose of Si ions (to create $\ll 1$ SiV on average) and optically verify whether a SiV resulted after annealing. Because of the ability to select implantation sites individually, the FIB process allows for such repeated low-yield implantation steps conditionally halted on the creation of the desired emitter number. An alternative approach to create precisely one quantum emitter is to implant only one ion at a time, as was recently demonstrated[44], combined with electron irradiation or co-implantation of other ion species to create vacancies[34] to drive the SiV conversion yield to unity.

The linewidths of the SiVs were measured in areas with 2.5 SiVs on average, distributed within only $\sim 55.4$ nm (FWHM) diameter, corresponding to an implantation dose of $\sim 10^{12}$ cm$^{-2}$, indicating that high densities of implanted SiVs are

not detrimental for their optical properties. Although we found that FIB-implanted SiVs are similar in homogeneous transition linewidth to 'natural', as-grown centres, the inhomogeneous linewidth of $\sim 51$ GHz (after 1,050 °C annealing) is still slightly broader than the $\sim 15$ GHz demonstrated for a similar SiV creation method with annealing temperatures around 1,200 °C (ref. 40). Potential causes include di-vacancy break down caused by higher temperatures, or near-surface strain and defects in the diamond due to polishing, which can be reduced by etching the damaged layer before implantation[45].

In summary, we demonstrated SiV creation with high spatial accuracy by FIB implantation of Si ions into bulk and nanostructured diamond. The SiV-positioning accuracy relative to the targeted nanocavity mode-maximum was $48 \pm 21$ nm, which is sufficiently precise to locate the SiV within $\sim 90\%$ of the mode-field intensity maximum of nanocavities or waveguides. We also demonstrated that the SiV creation yield can be increased after implantation by a factor of 10, up to 20%. The targeted implantation technique demonstrated here likely applies to other quantum emitters like the germanium defect centre in diamond[46] and other materials of interest, such as silicon carbide[47] or molybdenum disulfide; this would be particularly advantageous for materials where traditional nanofabricated masking is challenging.

The ZPLs of SiVs created by our method have optical linewidths within a factor of 1.4 of the lifetime limit, making them as narrow as naturally occurring SiVs described to date. Considering both this narrow linewidth and the narrow inhomogeneous distribution of implanted SiV of only $\sim 51$ GHz, this fabrication method represents a significant step towards the high-yield generation of thousands to millions of efficiently waveguide-coupled indistinguishable single-photon sources. Such arrays of atom-like quantum emitters would be of great utility for a range of proposed quantum technologies, including quantum networks and modular quantum computing[48,49], linear optics quantum computing[50,51], all-photonic quantum repeaters[52,53] and photonic Boson sampling[54].

## Methods

**Silicon ion implantation.** Focused ion implantation was performed at the Ion Beam Laboratory at Sandia National Laboratories using the nanoImplanter (nI). The nI is a 100 kV FIB machine (A&D FIB100nI) making use of a three-lens system designed for high mass resolution, using an ExB filter, and single ion implantation, using fast beam blanking. The ExB mass-filter (M/ΔM of $\sim 61$) separates out different ionic species and charge states from liquid metal alloy ion sources, providing the capability for implantation of $\sim 1/3$ the periodic table over a range of energies from 10 to 200 keV. For the Si implantation discussed here, we used an AuSbSi liquid metal alloy ion source with typical Si beam currents ranging from 0.4 to 1 pA. Fast beam blanking allows direct control over the number of implanted ions. We determine the number of implanted ions by measuring the beam current and setting the pulse length to target a given number of ions per pulse. The nI is a direct write lithography platform that uses electrostatic draw deflectors, controlled by a Raith Elphy Plus pattern generator, to position the beam. Single ion positioning is limited by the beam spot size on target. With typical spot sizes ranging from 10 to 50 nm, we have measured the targeting accuracy to be <35 nm for 200 keV Si++ beam using a series of ion beam-induced charge measurements.

For targeting into nanostructures, we align the ion beam relative to the sample by registering a secondary-electron image of the alignment markers generated using the ion beam to scan the sample. Shift, rotation and magnification corrections are calculated and applied in the pattern generator control package. This allows for any location within the write field to be individually targeted for implantation.

The lithography pattern is the original design file that was used to pattern the diamond thin film via electron beam lithography and reactive ion etching. Errors resulting from inaccuracy during electron beam lithography were not taken into account.

To create a single SiV per cavity with high probability, we implanted $\sim 20$ Si ions per cavity mode-maximum, yielding $\sim 1.8$ SiVs per cavity on average according to an extrapolated conversion efficiency of $\sim 3\%$ under Poisson statistics for 160 keVSi ions (Fig. 3) that target the middle of membrane at 106 nm.

**SiV creation and sample preparation.** We annealed the sample at 1,050 °C under high vacuum ($<10^{-6}$ mbar at maximum temperature) for 2 h to form SiV centres and eliminate other vacancy-related defects. Finally, we clean the sample surface through boiling tri-acid treatment (1:1:1 nitric:perchloric:sulfuric) and subsequent dry oxidation in a 30% oxygen atmosphere at 450 °C for 4 h.

**Room temperature measurement set-up.** We used a modified fluorescence microscope (Zeiss Axio Observer), customized to allow confocal illumination at 532 nm (Coherent Verdi) and single-mode fibre fluorescence collection. Collected fluorescence is spectrally filtered (Thorlabs FEL0650) and detected on avalanche photodiodes with single-photon resolution (Excelitas) or spectrally resolved on a grating spectrometer (Princeton Instruments, Acton SP2500i).

**Cryogenic measurement set-up.** These measurements were performed at 18 K in a closed cycle helium cryostat (Janis). A home-built confocal microscope collects the fluorescence with a high NA objective (Olympus UMplanfl 100 × 0.95 NA) and directs the emission to either the input of a single-mode fibre connected to an avalanche photodiode or to a free-space spectrometer with a resolution of ~61 pm (~34 GHz) at 737 nm (Princeton Instruments, IsoPlane SCT 320).

Photoluminescence excitation measurements were performed using a modified helium flow probe-station (Desert Cryogenics model PS-100) with a 0.95 NA microscope objective (Nikon CFI LU Plan Apo Epi 100 ×) inside the vacuum chamber. Details of this set-up are described in ref. 40.

**Analysis of spatial positioning precision.** To determine the spatial precision of the SiV implantation, we created and imaged a square array of SiV colour centres following the procedures in Methods A and D. We then fitted each SiV site with a 2D Gaussian to determine the location of the SiV centres below the diffraction limit, and considered only SiV sites with fluorescence intensities consistent with single emitters. Using these locations, we fitted a 2D grid allowing for affine transformation and found the distance between each SiV site and its nearest grid point. Finally, we binned the distances and fitted to a central $\chi$-distribution with two degrees of freedom (Rayleigh distribution), which describes the distribution of the distance $R = \sqrt{X^2 + Y^2}$ where $X$ and $Y$ are independent zero mean normal random variables with identical variance (Fig. 2b). The reported separation is the mean of the fitted $\chi$-distrbution corresponding to the mean separation in R (40 nm), and the error is the square root of the variance (20 nm). The mean separation in the $X$ and $Y$ directions is 0 nm with a s.d. of 32 nm.

**Analysis of targeted implantation accuracy.** To determine the positioning accuracy of the cavity-targeted SiV creation, we performed a spectrally resolved photoluminescence confocal scan at room temperature. At each pixel of a 2D 532 nm laser scan we recorded a spectrum and determined the intensity for different spectral positions. For each wavelength, we then plotted its 2D-intensity map as in Fig. 6d,e. This measurement allows comparison between the photonic crystal location, determined by Raman scattering of the 532 nm laser pump from the diamond (572.52 nm), which is present in the cavity region but not in the surrounding air holes, and the SiV location determined from the emitted fluorescence (at 736.98 nm). By fitting the measured emission patterns to 2d Gaussians, we estimate a relative positioning accuracy of 48(21) nm. The error is estimated from the 68% fitting confidence interval, which corresponds to one s.d.

**Data availability.** The data that support the findings of this study are available from the corresponding author.

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

## Acknowledgements

We thank Daniel L. Perry for his assistance in performing the implantation. This research was supported in part by the Army Research Laboratory Center for Distributed Quantum Information (CDQI), the CUA, the NSF, the AFOSR MURI, the Center for Integrated Quantum Materials (NSF grant DMR-1231319) and the DARPA QuINESS programme. Device fabrication was completed via controlled ion implantation with support from the Laboratory Directed Research and Development Program and the Center for Integrated Nanotechnologies, an Office of Science (SC) user facility at Sandia National Laboratories operated for the DOE (contract DE-AC04-94AL85000) by Sandia Corporation, a Lockheed Martin subsidiary.

## Author contributions

T.S., M.E.T. and M.W. carried out post-implantation sample processing, performed optical measurements and analysed the data. L.L. and J.Z. fabricated the diamond cavity sample. M.S. assisted with conversion yield experiments. J.L.P. and R.M.C. performed silicon FIB implantation. A.S., R.E.E., D.D.S. and C.T.N. performed the photoluminescence excitation measurement and the electron co-implantation experiment. T.S., M.E.T., M.W. and D.E. wrote the paper. E.S.B., M.D.L. and D.E. conceived and directed the project. All authors discussed the results and commented on the manuscript.
