## [Peer review file · Nature Communications]

Reviewers' comments:

Reviewer #1 (Remarks to the Author):

This paper presents a method of creation of SiV centers in diamond using focused ion beam (FIB). SiV centers are an important type of color centers in diamond for their potential applications in quantum information, especially in quantum interface between stationary and flying qubits, a key element in quantum networks. The authors demonstrated spatially precise implantation of SiV (with 10s of nm lateral uncertainty) and in particular demonstrate that the implanted SiV has broadening nearly limited by the photon emission lifetime, with the linewidth of a single SiV at low temperature (<13 K) the same as that of natural SiV, which is the best known result. They author realized implantation of SiV in optimal coupling positions inside nano-photonic structures (photonic crystal cavities), which will be very useful for cavity QED studies of such centers. Th results presented in this paper are of timely interest and significant for solid-state quantum information science. The paper is well written. I recommend publication of the paper with only some suggestions of minor revisions:

1. In caption of Fig. 1b, it is stated that the maxima are indicated by dashed lines in the figure. However, I do not see such dashed lines in the figure.
2. In Figure 2, the line colors "red" and "blue" are incorrectly labelled.
3. In Figure 2a, the SiV fluorescence and the overlaid image are hardly distinguishable.
4. What is the average number of SiV in each spot in Fig. 2a? What is the percentage of spots with single SiV?

Reviewer #2 (Remarks to the Author):

The paper “Scalable Focused Ion Beam Creation of Nearly Lifetime-Limited Single Quantum Emitters in Diamond Nanostructures” describes the creation and characterization of SiV in a photonic crystal using a custom build FIB system. The paper is well organized and described important results. However there are some comments to be considered:

1. The last sentence of the abstract is written ...This demonstration of deterministic creation of optical coherent solid state single quantum systems.....

The implantation of Si –ions was not deterministic and the creation yield is only 20 %. So the paper is far away from a deterministic creation of quantum objects.

2. First sentence second paragraph: Here, we introduce a new method for positioning... The implantation method using a FIB is not really new. Implantation with a FIB system is done since several decades by a large number of groups like Shinada, Wieck etc, also the implantation in Diamond is already performed by Lesik et al. using Nitrogen [your citation 30].

3. Page 2, 2. Paragraph: AFM tip does not require modification of the fabrication process..... A comparison between FIB and pierced AFM-tip is difficult. Both techniques have several pros and cons. However an AFM tip is usually run in non contact mode thus the collision can be avoided and it makes less effort to build a pierced AFM tip in comparison to setup a LMIS for Silicon ions.

4. Page 3 1. Paragraph: ---High-purity chemical vapor deposition (CVD) similar sentence twice. The whole paragraph and the following description of Fig.2 are very confused.

5. Page 4 1. Paragraph: .four spots with silicon ions in increasing dose.....What kind of energy is used for this implantations?

6. Fig. 4: Between 5000 and 10000 ions the luminosity is expected to increase by a factor of 2, but this seems not the case?

7. Page 4 End of 1. paragraph: That the conversion efficiency of focused ion beam implantation is limited by the vacancy density in the diamond..... In higher depth the conversion yield is usually much higher. But the ions decelerated thus the number of vacancies in diffusion distance to the implanted ion is more or less similar for deep and shallow implanted ions. One explanation could be a change of the Fermi level at the surface and an influence of the creation process and/or the charge state.

I miss an experiment that proves the relation between numbers of SiVs and luminosity of SiVs.

Additionally is there an experimental proof that the increase of the luminosity is related to an

increase of SiV centers only and not effected by other factors e.g. change of luminosity of the individual centers.

Reviewer #3 (Remarks to the Author):

Title: “Scalable focused ion beam creation of nearly lifetime-limited single quantum emitters in diamond nanostructures”

The authors use a modified (silicon) focused ion beam to implant/dope single crystal diamond with silicon-vacancy SiV- color centers and investigate their properties for quantum networking at scale. Characteristics of the implant method are: (i) fast raster scanning over large substrates, (ii) flexible ion choice, (iii) and controllable dosing. The color center chosen for this work (SiV- in diamond) is now well-known from other studies to possess an inversion symmetry (zero dipole moment) property that makes atomic transitions robust to e.g. materials processing of quantum emitters near diamond surfaces as required by many quantum technologies.

There are no big surprises here. The combination of FIB implant into pre-fabricated diamond cavities is shown in another recent paper (Lukin et al Science 18 Nov 2016: Vol 354, Issue 6314, pp. 847-850) co-authored by many of the authors (Sandia & Harvard) of this particular work. Otherwise, the cycle Implant → Anneal → Test & Measure → Repeat → Publish is now well-established to the point where there is no significant need to present here other citations from the diamond NV center literature. It is not totally clear to me that this work is sufficiently strongly and positively differentiated from the Science publication to warrant additional publication in Nature Communications. That being said, the work appears to be a sound job and another step in the development of scalable quantum systems based on diamond and other solid-state hosts.

We of course know that there are many color centers with different properties to be determined. Recently a germanium GeV- color center in diamond is shown to have essentially identical level scheme and crucial inversion symmetry though with different transition frequency. The authors are no doubt working on this system in parallel. Meanwhile, it seems easy to imagine that there are other color centers exhibiting inversion symmetry and zero Stark shift to be realized in diamond and also other hosts. At this point, rather than claiming “Mission Accomplished” as in the title of this paper it is perhaps more intriguing to try to take a first-principles approach to materials engineering identical quantum emitters. For example, is the more massive Ge (vs. Si) dopant more or less favorable to the inhomogeneous line distribution via ion implantation? You might argue naively either way. How about atom-atom bond length in the semiconductor host? Etc. While such questions are obviously out of the scope of this work, diamond GeV- already exists and is in the zeitgeist and you could probably do the same experiment with it tomorrow by the close of business and submit the next paper by the next week. So some kind of concluding comment regarding a look to the future of the quantum materials engineering seems appropriate.

Response to the Reviewers

Dear Reviewers,

We are thankful for your careful consideration of our manuscript # NCOMMS-16-25606 with the title “Scalable Focused Ion Beam Creation of Nearly Lifetime-Limited Single Quantum Emitters in Diamond Nanostructures”. We have carefully revised the manuscript based on your comments and suggestions, as detailed below.

We have copied the reviewer’s comments and have colored the font of our response in blue.

Response to Reviewer #1

This paper presents a method of creation of SiV centers in diamond using focused ion beam (FIB). SiV centers are an important type of color centers in diamond for their potential applications in quantum information, especially in quantum interface between stationary and flying qubits, a key element in quantum networks. The authors demonstrated spatially precise implantation of SiV (with 10s of nm lateral uncertainty) and in particular demonstrate that the implanted SiV has broadening nearly limited by the photon emission lifetime, with the linewidth of a single SiV at low temperature (<13 K) the same as that of natural SiV, which is the best known result. They author realized implantation of SiV in optimal coupling positions inside nano-phonic structures (phonic crystal cavities), which will be very useful for cavity QED studies of such centers. Th results presented in this paper are of timely interest and significant for solid-state quantum information science. The paper is well written. I recommend publication of the paper with only some suggestions of minor revisions:

1. In caption of Fig. 1b, it is stated that the maxima are indicated by dashed lines in the figure. However, I do not see such dashed lines in the figure.

We thank the reviewer for pointing out this error. We have revised the caption, which now reads:

‘Intensity distribution of the fundamental L3 cavity mode with three Si target positions: the three mode maxima along the center of the cavity are indicated by the dashed circle. The central mode peak is the global maximum.’

2. In Figure 2, the line colors "red" and "blue" are incorrectly labelled.

We thank the reviewer for pointing out this error. It has been corrected in the revised manuscript:

The Figure 2b and c captions have been modified to ‘Blue curve: fit to Rayleigh distribution.’ and ‘Red points indicate data (without background subtraction), and the blue line is a fit to the function ...’

As well as: ‘...while the blue dashed lines indicate the 95% confidence interval on the fit. d) Ensemble (black) and single-emitter...’

3. In Figure 2a, the SiV fluorescence and the overlaid image are hardly distinguishable.

The red circles resemble the area within we expect the fluorescent SiV to be located. Therefore, the circle 'has' to be a small circle which partially covers the SiV fluorescence. However, comparing the fluorescence maximum for the circle shows visually (only qualitatively) that the SiV are well distributed within the circles.

We adjusted the linewidth to help the visibility.

4. What is the average number of SiV in each spot in Fig. 2a? What is the percentage of spots with single SiV?

The SiV occurrence in Fig. 2a is randomly distributed and follows a Poissonian distribution. We implanted 50 ions/spot, so we expected a Poisson parameter of ~ 1 with 2% yield. We identified 107 spots consistent with single-photon emission PL out of ~ 320 sites, with 41 sites identified as having higher than single-photon PL. This corresponds to a Poisson parameter of ~ 0.7 . We thank the reviewer for bringing up this point, as in collecting this data we realized that the histogram displayed in Figure 2b contains all SiV points regardless of their single-emitter PL.

We have updated Figure 2b to only contain those sites with single-emitter level PL, and we note that the parameter describing the distribution is unchanged with the level of precision quoted in the text (mean separation of 39.63 previously vs. 39.76 updated, hence about 40 nm).

Response to Reviewer #2

The paper "Scalable Focused Ion Beam Creation of Nearly Lifetime-Limited Single Quantum Emitters in Diamond Nanostructures" describes the creation and characterization of SiV in a photonic crystal using a custom build FIB system. The paper is well organized and described important results. However there are some comments to be considered:

1. The last sentence of the abstract is written ...This demonstration of deterministic creation of optical coherent solid state single quantum systems.....
The implantation of Si⁻ions was not deterministic and the creation yield is only 20 %. So the paper is far away from a deterministic creation of quantum objects.

The reviewer is correct in that the creation of SiV is not deterministic. Therefore, we change our wording to "high-yield, targeted creation", but argue the following.

Throughout the abstract and the manuscript, we refer to 'system' as a nanostructure-SiV system, and not the SiV alone. Furthermore, as emitters can be addressed in the spectral domain (if separated $>$ emission linewidth, which is very likely for an inhomogeneous distribution of 51GHz), there is no need to have exactly one emitter per device to have nearly 100% device yield. In other words, if $p(n)$ is the probability to create n SiV centers, the device creation yield is

not $p(1)$ but $1-p(0)$. Even if one assumes that more than 5 SiVs are unacceptable, the device creation yield is then about $1-p(0)-p(n>5)$, which can in general be quite large. Therefore, describing the fabrication process as nearly deterministic is justified.

We change:

“This demonstration of deterministic creation of optically coherent solid-state single quantum systems”

to

“This demonstration of high-yield, targeted creation of optically coherent solid-state single quantum systems”

2. First sentence second paragraph: Here, we introduce a new method for positioning... The implantation method using a FIB is not really new. Implantation with a FIB system is done since several decades by a large number of groups like Shinada, Wieck etc, also the implantation in Diamond is already performed by Lesik et al. using Nitrogen [your citation 30].

We agree with the reviewer that implantation has been carried out by a large number of groups. However, we argue, that (maskless) ion-beam based nanostructure-specific implantation is a new method for the post-fabrication creation of SiV in a large number of e.g. nanocavities. Previous maskless ion-beam based defect center creation did not target nanostructures, but only site-specific implantation in bulk diamond (Ref. 30). Our method does make the fabrication of SiV-cavity systems, compared to previous, implantation-mask based methods, much less complex and feasible. Furthermore, implantation through apertures can lead to increased vertical straggling and ions could be decelerated resulting in larger implantation depth uncertainty.

We agree with the reviewer, that a Si-specific FIB tool is a demanding machine, but given its accessibility and the disadvantages of masked targeting, we argue that our fabrication method is a very important step for reliably making emitter-nanostructure systems.

We change:

“Here, we introduce a new method for positioning emitters relative”

to

“Here, we introduce a method for the scalable positioning of SiV emitters relative”

3. Page 2, 2. Paragraph: AFM tip does not require modification of the fabrication process..... A comparison between FIB and pierced AFM-tip is difficult. Both techniques have several pros and cons. However an AFM tip is usually run in non contact mode thus the collision can be avoided and it makes less effort to build a pierced AFM tip in comparison to setup a LMIS for Silicon ions.

We agree with the reviewer that a comparison of these two different methods is challenging. However, these methods achieve the same goal: targeted implantation without the need for fabricating a mask on top of the sample (which is highly relevant if a certain surface termination is required), therefore a comparison is required. Furthermore, in our discussion, we do acknowledge this work but also point to its disadvantages compared to our method.

Evaluating the complexity of building either a pierced AFM system (which has to meet the stringent requirements inside an ion implanter) or a Si-specific FIB is out of the scope of this introduction and in our understanding not relevant for the evaluation of our method. The discussed advantages of the FIB method well warrant its implementation for the scalable generation of SiV-cavity systems and therefore also its detailed introduction as a method with several advantages.

4. Page 3 1. Paragraph: ---High-purity chemical vapor deposition (CVD) similar sentence twice. The whole paragraph and the following description of Fig.2 are very confused.

We thank the reviewer for pointing out this repetition. We have deleted the following sentence from the manuscript:

“We applied commercially available high-purity chemical vapor deposition (CVD) diamond substrates (Element6).”

5. Page 4 1. Paragraph: .four spots with silicon ions in increasing dose.....What kind of energy is used for this implantations?

We thank the reviewer for pointing at this missing information. The Si ions were implanted with 100 keV corresponding to an implantation depth of about 68 nm.

We have added this information in the text.

6. Fig. 4: Between 5000 and 10000 ions the luminosity is expected to increase by a factor of 2, but this seems not the case?

We apologize for the insufficient image quality and thank the reviewer for noting this issue. The reviewer is right, the luminosity does increase between the 5000 and 10000 ion cases by a factor of two. Although this is clear in our version of the manuscript, we have adjusted the contrast/colormap slightly to make the difference more visible.

7. Page 4 End of 1. paragraph: That the conversion efficiency of focused ion beam implantation is limited by the vacancy density in the diamond..... In higher depth the conversion yield is usually much higher. But the ions decelerated thus the number of vacancies in diffusion distance to the implanted ion is more or less similar for deep and shallow implanted ions. One explanation could be a change of the Fermi level at the surface and an influence of the creation process and/or the charge state.

We agree with the reviewer that there is a complex interplay between the number of vacancies, the number of defects, number of ions, the distance to the surface (possibly due to a modified Fermi level), and many more. It is however, not the scope of this article to evaluate the depth-dependent conversion efficiency and yield. We do not quantify the relation between implantation dose and energy to number of vacancies and created SiV but only make a qualitative statement, sufficient to support our (and previous) experimental observation that the implantation parameters influence the SiV creation yield.

I miss an experiment that proves the relation between numbers of SiVs and luminosity of SiVs.

Additionally is there an experimental proof that the increase of the luminosity is related to an increase of SiV centers only and not effected by other factors e.g. change of luminosity of the individual centers.

We thank the reviewer for pointing a this critical property. In the manuscript, we report in several paragraphs and related images the relation between number of implanted Si ions to luminescence as well as the number of SiVs to luminescence. In particular, we report typical 30k counts/s per single SiV as determined by autocorrelation measurements. Taking into account the background, this gives about 19k counts/s of SiV photons. We use this number to estimate the conversion yield in Fig. 3. This data also shows that we determine a dose and energy dependent luminosity and creation yield. The latter we attribute to a complex interplay of created defects and implanted Si. We discuss two different mechanisms leading to energy and dose dependent yield, (i) an accumulation of charged defects in the diamond lattice that lead to ionization, and (ii) reduced yield resulting from lattice damage that accumulates in the form of multivacancy defects as the diamond lattice approaches the graphitization threshold. The interpretation of our findings and corresponding literature [34,35] led us to the conclusion that we can exclude a change of luminosity of individual centers. The reviewer is right this has not been finally proven as it requires knowledge of the exact number of SiV which is difficult to estimate (excluding estimation by luminosity). Even with e.g. energy-dispersive X-ray spectroscopy (EDS) it is difficult to distinguish between Si defects and specific Si-vacancy defects. Therefore we rely on our luminosity data which is strongly supported by the literature and our own experiments.

Response to Reviewer #3

Title: “Scalable focused ion beam creation of nearly lifetime-limited single quantum emitter in diamond nanostructures” The authors use a modified (silicon) focused ion beam to implant/dope single crystal diamond with silicon-vacancy SiV- color centers and investigate their properties for quantum networking at scale. Characteristics of the implant method are: (i) fast raster scanning over large substrates, (ii) flexible ion choice, (iii) and controllable dosing. The color center chosen for this work (SiV- in diamond) is now well-known from other studies to possess an inversion symmetry (zero dipole moment) property that makes atomic transitions robust to e.g. materials processing of quantum emitters near diamond surfaces as required by many quantum technologies.

There are no big surprises here. The combination of FIB implant into pre-fabricated diamond cavities is shown in another recent paper (Lukin et al Science 18 Nov 2016: Vol 354, Issue 6314, pp. 847-850) co-authored by many of the authors (Sandia & Harvard) of this particular work. Otherwise, the cycle Implant Anneal Test & Measure Repeat Publish is now well-established to the point where there is no significant need to present here other citations from the diamond NV center literature. It is not totally clear to me that this work is sufficiently strongly and positively differentiated from the Science publication to warrant additional publication in Nature

Communications. That being said, the work appears to be a sound job and another step in the development of scalable quantum systems based on diamond and other solid-state hosts.

We thank the reviewer for the positive comments and would like to point out the differences to the recent Science article (Science Vol 354, pp. 847-850). The Science article demonstrates and characterizes a quantum nanophotonic device whose fabrication was enabled by the technology described in this manuscript, highlighting the power of this technique and its potential application to the development of quantum optical networks.

The submitted manuscript complements and extends the previous work, in particular the performance, experimental details, and further quantitative analysis of the FIB implantation and SiV creation process. In the submitted manuscript, we characterize the FIB emitter-creation method in terms of spatial accuracy and precision as well as resulting emitter spectral properties. We additionally supply an important method for the stepwise increase of conversion yield towards unity yield. Finally, we show the application to 2-dimensional photonic crystal cavities that have several advantages over 1-d cavities.

We believe that this work is an important advance in the state-of-the-art of quantum emitter fabrication independent of the device shown in the Science article, and provides a quantitative baseline for future investigations in this direction.

We of course know that there are many color centers with different properties to be determined. Recently a germanium GeV- color center in diamond is shown to have essentially identical level scheme and crucial inversion symmetry though with different transition frequency. The authors are no doubt working on this system in parallel. Meanwhile, it seems easy to imagine that there are other color centers exhibiting inversion symmetry and zero Stark shift to be realized in diamond and also other hosts. At this point, rather than claiming “Mission Accomplished” as in the title of this paper it is perhaps more intriguing to try to take a first-principles approach to materials engineering identical quantum emitters. For example, is the more massive Ge (vs. Si) dopant more or less favorable to the inhomogeneous line distribution via ion implantation? You might argue naively either way. How about atom-atom bond length in the semiconductor host? Etc. While such questions are obviously out of the scope of this work, diamond GeV- already exists and is in the zeitgeist and you could probably do the same experiment with it tomorrow by the close of business and submit the next paper by the next week. So some kind of concluding comment regarding a look to the future of the quantum materials engineering seems appropriate.

We agree with the reviewer that there are many potential hosts and implantable species, and that this work explores only one particular combination that happens to have favorable spectral properties. We also share the reviewers attitude that this work, having answered the initial question of how to precisely place a quantum emitter, now allows us to ask follow-on questions regarding what species should be placed, in what host, to what end. This work will be extended with other materials, other systems, and by other researchers.

Within the scope of this paper, however, we have tried to limit our speculation and only discuss emitter engineering as it directly relates to FIB implantation as well as creation of an already well-explored and suitable quantum emitter, in contrast to other methods, e.g., un-focused implantation, masked implantation, or doping in a growth process, and other emitters, e.g., GeV,

respectively. The advantage given by this technique is that it allows positioning with great accuracy and therefore coupling of emitter degrees of freedom to other systems with high probability, without sacrificing quality and optical properties of the emitter.

Building on this, we expect the future of quantum materials engineering to move beyond the bare properties of single emitters to treat coupled systems, be they photonic, phononic, or beyond, as the core element. Our method could be an enabler of such coupled systems based on solid state quantum defects that can be created by FIB implantation.

We thank the reviewer for these stimulating comments and have updated the conclusion to reflect this discussion.

REVIEWERS' COMMENTS:

Reviewer #2 (Remarks to the Author):

Dear Author

I accept the revised paper for publication.